# Treatment Targets in Ulcerative Colitis: Is It Time for All In, including Histology?

**DOI:** 10.3390/jcm10235551

**Published:** 2021-11-26

**Authors:** Panu Wetwittayakhlang, Livia Lontai, Lorant Gonczi, Petra A. Golovics, Gustavo Drügg Hahn, Talat Bessissow, Peter L. Lakatos

**Affiliations:** 1Division of Gastroenterology, McGill University Health Center, Montreal, QC H3G 1A4, Canada or wet.panu@gmail.com (P.W.); golovics.petra@gmail.com (P.A.G.); gustavo.hahn@mail.mcgill.ca (G.D.H.); talat.bessissow@gmail.com (T.B.); 2Gastroenterology and Hepatology Unit, Division of Internal Medicine, Faculty of Medicine, Prince of Songkla University, Songkhla 90110, Thailand; 3First Department of Medicine, Semmelweis University, H-1083 Budapest, Hungary; livia.lontai@gmail.com (L.L.); lorantgonczi@gmail.com (L.G.); 4Department of Gastroenterology, Hungarian Defence Forces, Medical Centre, H-1062 Budapest, Hungary; 5Graduate Course Sciences in Gastroenterology and Hepatology, School of Medicine, Universidade Federal do Rio Grande do Sul, Porto Alegre 90035-002, Brazil

**Keywords:** treat to target, treatment target, ulcerative colitis, mucosal healing, inflammatory bowel diseases, histological remission

## Abstract

The main therapeutic goal of ulcerative colitis (UC) is to induce and maintain remission to prevent long-term disease progression. Treat-to-target strategies, first introduced by the STRIDE consensus and updated in 2021, have shifted focus from symptomatic control toward more stringent objective endpoints. Today, patient monitoring should be based on a combination of biomarkers and clinical scores, while patient-reported outcomes could be used as short-term targets in monitoring disease activity and therapeutic response. In addition, endoscopic healing was the preferred long-term goal in UC. A Mayo endoscopic score (MES) ≤ 1 can be recommended as a minimum target. However, recent evidence suggests that more stringent endoscopic goals (MES of 0) are associated with superior outcomes. Recently, emerging data support that histological remission (HR) is a superior prognostic factor to endoscopic healing in predicting long-term remission. Despite not yet being recommended as a target, HR may become an important potential therapeutic goal in UC. However, it remains questionable if histological healing should be used as a routine assessment in addition to clinical, biomarker, and endoscopic targets in all patients. Therefore, in this review, our aim was to discuss the current evidence for the different treatment targets and their value in everyday clinical practice.

## 1. Introduction

Ulcerative colitis (UC) is a chronic inflammatory and progressive disease with relapsing episodes. The course of the disease varies from asymptomatic, mild to extensive inflammation of the colon, resulting in disability, intestinal damage, permanent fibrosis, and the need for surgery [1]. Approximately 10–20% of UC patients have an aggressive course, while another 30% of patients present with disease extension, where the cumulative risk of relapse is 70–80% at 10 years. Almost 50% of patients require hospitalization, and the 10 year cumulative risk of colectomy is 10–15% [2,3,4,5,6]. It is important to note that approximately 50% of UC patients will not have a progressive disease course [7]. Thus, early stratification and early identification of patients with risk for disease progression and/or severe disease are the key to treatment in UC. Traditional therapeutic approaches target the control of symptoms, do not necessarily alter the natural course of the disease, and can lead to delays in achieving endoscopic healing or the development of UC-related complications, including hospitalizations, colectomy, and colorectal cancer, especially in patients with moderate to severe disease [8]. The widely available biologics and small molecules have introduced more opportunities to improve disease outcomes and increase the ability to reach beyond the conventional therapeutic goals.

The concept of the treat-to-target approach, which was first put forward by the Selecting Therapeutic Targets in Inflammatory Bowel Disease (STRIDE) consensus in 2015, aims to achieve disease remission by adjusting therapy according to the achievement of treatment targets [9]. It has shifted the goal of UC treatment from only symptomatic control to support targeting objective therapeutic endpoints to prevent long-term disease complications. In 2021, STRIDE-II was updated, encompassing evidence- and consensus-based recommendations for treat-to-target strategies [10]. Symptomatic relief and normalization of serum and fecal biomarkers have been determined as short-term targets. The primary therapeutic target in both UC and CD should be the composite endpoint of both clinical/patient-reported outcomes (PROs) and endoscopic remission. For UC, clinical/PRO remission was defined as resolution of rectal bleeding and diarrhea/altered bowel habits, with endoscopic remission described as MES of 0–1. Despite histologic remission not being recommended as a treatment target in STRIDE II, recent data have shown the benefit of histological remission (HR) in the long-term outcome [11]. HR has recently been recognized as an important prognostic factor and potential treatment endpoint in patients with UC in clinical studies.

Multiple studies support the benefit of treat-to-target strategies; yet, in UC, the picture is more subtle since a proportion of patients may be sufficiently maintained with a conventional therapeutic approach. This review aims to present and discuss existing evidence on treatment targets and monitoring strategies, including clinical, endoscopic, biomarker, and particularly histological targets in every clinical practice of UC management. We highlight the important state-of-the-art care and practical up-to-date evidence in the different treatment targets.

## 2. Clinical Targets

Until recently, clinical remission remained the main target of treatment in UC. Clinical scoring systems based on symptoms have been used to estimate disease activity in IBD management for a long time. Evaluating disease activity is crucial for optimizing medical management to reach predefined treatment targets [9]. Although endoscopic assessment provides objective information about the mucosal healing, it is invasive and comes with an additional cost. There has been an aim to develop clinical scoring systems, which have a strong correlation with endoscopic activity and can monitor disease activity without repeated endoscopic evaluations.

The Mayo score and the partial Mayo score (pMayo) are the most frequently used clinical disease activity scores in UC clinical trials and clinical practice [12]. The Mayo score is a composite tool, including four variables (stool frequency, rectal bleeding, physician’s global assessment, and endoscopic evaluation) ranging from 0–12 points. Clinical response is defined as a decrease of ≥3 points, and clinical remission is defined as a total Mayo score ≤ 2 with no sub-score >1. The pMayo score is not included in endoscopic evaluation, on a scale of nine points. Of note, clinical remission by the Mayo scores allows streaks of blood in the stool, which cannot be considered as complete clinical remission [12,13,14]. The Mayo scores include a combination of clinical and endoscopic measures. Thus, endoscopy is part of the routine patient assessment in UC even with the traditional approach, likely due to the need for flexible sigmoidoscopy only rather than a full ileocolonoscopy, if cancer surveillance is not necessary.

The Simple Clinical Colitis Activity Index (SCCAI) is another known clinical score that includes six variables (frequency of bowel movements during the day and night, urgency of defecation, blood in the stool, general wellbeing, and extraintestinal manifestations), which correlated well with Mayo score and accurately predicted disease activity in validating studies [15,16]. Clinical response is defined as an SCCAI decrease ≥ 2 points from baseline, while clinical remission is defined as a total SCCAI score ≤ 2 points [17].

In the last few years, patients’ perspectives on disease activity have become an essential assessment tool and treatment target [18]. Thus, patient-reported outcomes (PROs) or PRO2 are composed of two-component stool frequency (SF) and rectal bleeding (RB), which have become the current standard target for assessing clinical symptoms in UC in clinical trials. The resolution of diarrhea and the absence of bloody stool combined are independent predictors of relapse-free, colectomy-free survival and long-term outcomes in UC [19]. Complete clinical remission with PRO2 was associated with endoscopic healing (MES of 0/1) in approximately 80–90% of patients in a prospective study of infliximab-treated patients [13]. Consequently, normal SF and the resolution of RB are the main clinical targets in UC.

The updated STRIDE-II introduced criteria for defining clinical response using the PRO2, defined as a decrease of ≥50% in PRO2. Clinical remission is defined as RB and SF of 0. Clinical response should be an immediate (short-term) target, whereas clinical remission is an intermediate target [10]. However, knowing that SF normalization is highly unreliable, less stringent criteria, allowing SF 0/1, may be prudent to define clinical remission [20,21].

Recent studies have shown that there is a good correlation between PRO2 and endoscopic activity. Earlier studies compared RB, SF, and PRO2 with endoscopic remission in the induction of mesalamine. AUCs for RB alone, SF alone, and PRO2 were 0.78, 0.85, and 0.90 for endoscopic remission (MES ≤ 1) [22]. in later post hoc studies from the ULTRA 1 and 2 trials, at week 8 treatment, the positive predictive values (PPVs) for SF of 0, RB of 0, and PRO2 remission with MES ≤ 1 were as high as 69%, 84%, and 90%, respectively [23]. These post hoc analyses showed that the two-item PRO2 correlated well with endoscopic activity, and that the combined use of RB and SF was superior in predicting endoscopic remission.

In a systematic review, our group showed the PRO2 had a moderate to strong correlation with endoscopic activity. The absence of RB identified patients with an inactive disease with higher sensitivity than normalization SF [24]. A more recent meta-analysis of five studies, including 2132 patients with UC, found that an RB score of 0 identified patients in endoscopic remission with a pooled sensitivity value of 81% and a specificity of 68%. A combined RB and SF subscore of 0 identified patients in endoscopic remission with a pooled specificity value of 96% [25].

An even more recent prospective study by Golovics et al. evaluated the correlation between PROs and traditional clinical scores and endoscopy in 171 UC patients. RB of 0, SF of 0, PRO2 remission (RB = 0 and SF ≤ 1), partial Mayo (≤2), and SCCAI (≤2.5) remission were all similarly associated with mucosal healing defined by MES (0 or ≤1) or UCEIS (≤3) scores in ROC analysis (AUC: 0.93–0.72 in all cases). However, no clinically meaningful differences were found in accuracy across different clinical scores to predict endoscopic activity [26].

However, endoscopy—even sigmoidoscopy—is invasive, costly, and time-consuming; hence, the ability to predict endoscopic remission using clinical scores or PROs would benefit both patients and clinicians. PRO2 remission is likely to reflect endoscopic remission, especially when less stringent endoscopic criteria are used (MES ≤ 1; >90% PPV). In contrast, PRO2 remission may not differentiate reliably between complete mucosal healing and minor endoscopic activity [20,25,26]. In addition, approximately 20% of patients in clinical remission may still have significant endoscopic disease activity [27], and complete normalization of SF is not always observed in patients with endoscopic healing [23].

As a take-home message and practical conclusion of the above studies, rectal bleeding and stool frequency scores can, thus, reliably identify patients with an MES of 0–1. Nevertheless, it is suboptimal in differentiating between patients with an endoscopic score of 0 or 1. In clinical practice, this means that, after commencing systemic steroid and/or anti-TNF or biologic therapies in moderate to severe UC, evaluation of clinical response should be performed at 2–4 weeks, while evaluation of clinical remission should be performed at 4–6 weeks of treatment. Given the lack of high-quality data, the optimal timeframe for evaluating symptomatic response to various treatments is only a rough estimate of experts’ opinions. In mild UC, evaluation of clinical response may be assessed 12 weeks after initiation of treatment. In this review, we propose the recommended treatment targets in UC, shown in Table 1.

In summary, clinical response and remission, including PROs (normal stool frequency and absence of rectal bleeding), are the short-term treatment target in UC. In addition, clinical scores have a good correlation with endoscopic activity, which can be used to prioritize patients for endoscopic evaluation. However, clinical remission alone is insufficient to be used as a long-term treatment target in UC. Moreover, we may need different cutoffs/definitions of clinical remission/control for clinical trials and everyday practice.

## 3. Biomarker Targets

As repeated endoscopic evaluations are limited due to invasiveness and cost, biomarkers play an important role in assessing and monitoring of disease activity in UC. In clinical practice, serological markers, C-reactive protein (CRP), erythrocyte sedimentation rate (ESR), and fecal calprotectin (FC) are the most thoroughly studied biomarkers in IBD.

Although elevated CRP level shows better correlation with disease activity in CD and is not a very sensitive marker in UC in general, it has been identified as a valuable predictor of early treatment outcomes and steroid response in patients with acute severe UC (ASUC), as highlighted by the Oxford score (CRP > 45 mg/L) [28]. According to a recent study, serial measurements may be superior to fecal biomarkers for evaluating colon-wide active inflammation in patients with severe colitis [29]. Much fewer data support the applicability of CRP measurements in mild–moderate disease, as many patients with UC do not have elevated CRP levels. Approximately 50% of patients with active UC have normal CRP levels [30]. CRP with a lower cutoff (≤8 mg/L) has low sensitivities for detecting endoscopic remission (sensitivity 51–53%, specificity 69–71%), suggesting that CRP alone is not accurate to reflect endoscopic severity [31]. In addition, a meta-analysis of 19 studies showed that CRP has high specificity (0.92, 95% CI 0.72–0.96) for detecting active endoscopic disease, but sensitivity is poor (0.49, 95% CI 0.34–0.64), and a negative test does not reliably exclude the presence of active inflammation [32].

Fecal calprotectin (FC) is a biomarker of intestinal inflammation, a more sensitive marker than CRP for predicting mucosal healing in UC [33,34]. Several studies have shown that FC has a good correlation with endoscopic, histological activity, and relapse of UC [35,36,37,38,39]. A recent meta-analysis showed FC has a sensitivity of 78% and specificity of 79% with an AUC of 0.85 to predict endoscopic mucosal healing. Although the included studies used a wide range of cutoff values (14 to 251 μg/g) [40]. Identifying the optimal FC concentration cutoff values best predictive of disease activity is challenging. D’Haens et al. suggested a value of 250 μg/g, as levels above this concentration were associated with active mucosal disease (Mayo endoscopic score, MES > 0) in UC (sensitivity 71%, specificity 100%) [41]. A meta-analysis by Mosli et al. showed FC was more sensitive than CRP (88% vs. 49%) in patients with IBD and was more sensitive in UC than Crohn’s disease. In UC, pooled sensitivity and specificity for FC to predict endoscopically active inflammation were 0.88 (95% CI 0.84–0.92) and 0.79 (95% CI 0.68–0.87), respectively [32].

In IBD patients treated with anti-TNF, FC values were significantly lower in patients with clinical and endoscopic remissions compared to patients with only clinical remission (50 vs. 288 μg/g) [42]. Moreover, FC value < 100 μg/g can be used as a cutoff point for clinical remission in anti-TNF-treated patients [43]. Another study found an FC value > 250 μg/g to predict active endoscopic activity (MES > 0) with a sensitivity of 71% and a specificity of 100% (PPV 100%, NPV 47%) [41]. Moreover, a recent meta-analysis including 49 studies of patients with UC supported the use of FC in predicting endoscopic remission with a sensitivity of 78% and specificity of 79%, with the most common FC cutoff levels between 150 and 250 μg/g [40].

Two meta-analyses suggested that FC cutoff values of 50 μg/g can be used to screen for further endoscopy evaluation, with a sensitivity of 92% and a specificity of 60%. However, using this low cutoff value, 40% of endoscopy-negative patients will undergo an unnecessary invasive procedure. A cutoff value at 250 μg/g can be used as a confirmed test to contemplate escalating therapy with a pooled sensitivity of 80% and specificity of 82%. Using this value, 18% of those without active disease would be identified as false positives and receive excessive treatment [32,34].

However, one of the major limitations in real-life clinical practice of FC, a substantial number of patients with UC have FC values in the “intermediate or gray zone” (values between 100 and 250 μg/g), which hinder the interpretation of disease activity. Thus, the combination of biomarkers and clinical indices can add value for predicting disease activity. A cohort study by Bodelie et al. evaluated the additional value of this subgroup of UC patients with these indefinite FC values, showing that a combination score of biomarkers (FC 100–250 μg/g and CRP < 5 mg/L) and clinical activity indices (SCCI < 3) was able to identify endoscopic disease activity in UC with a sensitivity of 88% and specificity of 75% (PPV 93%, NPV 60%) [44].

In a post hoc analysis of patients with moderate to severe UC treated with biologic agents or small-molecule inhibitors, using FC in combination with clinical score (PRO2), the authors reported that patients with UC achieving RBS 0 and SFS 0–1, FC ≤ 50 (±10) μg/g have a low likelihood of moderate to severe active endoscopic inflammation and may avoid endoscopy with a false-negative rate of only 4.5%. In contrast, patients with RBS 2–3 and SFS 2–3, FC ≥ 250 (±20) μg/g, with a high likelihood of active endoscopic activity (MES 2–3), can also practically avoid endoscopy in with a false-positive rate < 5%. The greatest uncertainty in diagnostic performance for FC was observed in UC patients achieving RBS 0 but having SFS 2/3, where false-negative and false-positive rates were consistently > 10%, and endoscopic evaluation may be warranted [45].

Moreover, FC is useful to predict relapse of the disease in UC patients with remission, as first published by Tibble et al., who showed that among patients in remission with high FC levels using a cutoff value of 250 μg/g, almost 90% relapsed within the following 12 months [39]. Several studies have shown that increased concentrations of FC in patients with the quiescent disease can predict disease relapse within 1 year. These studies used various cutoff FC levels, ranging from 50 to 300 μg/g [30,35,46,47,48].

In a prospective study of patients under maintenance of anti-TNF treatment, FC levels quantified at 4 month intervals predicted relapse/remission over the following 4 months. Cutoff values to predict remission were 130 μg/g (negative predictive value of 100%) and 300 μg/g to predict relapse (PPV of 78%) [49]. Furthermore, after stopping anti-TNF therapy in patients in deep remission (clinical and endoscopic remission and baseline FC < 100 mg/g), FC seems to rise and remain elevated before clinical and endoscopic relapse [43]. A systematic review by Heida et al. found that an elevation of FC levels in patients with clinical remission was correlated with an increased probability of relapse from 53% to 83% within the next 2–3 months, whereas patients with repeatedly normal FC had a 67–94% probability of remaining in remission over that same period [48].

In a recent meta-analysis including 14 studies of 1110 participants with UC, nine studies used an FC cutoff ≥ 150 μg/g, which had a sensitivity of 71% (95% CI 65–78%) and specificity of 86% (95% CI 0.82–89%), whereas five studies used an FC cutoff < 150 μg/g, which gave a sensitivity of 79% (95% CI 71–85%) and specificity of 64% (95% CI 58–69%) for FC in predicting relapse. Most of the studies had a follow-up time ≥ 12 months [47].

The next level of control is histological remission. Multiple studies have shown a strong correlation between FC levels and histologic activity in patients with UC. FC can be used to predict histologic activity in UC, but the cutoff level varies across studies [36,37,50,51,52,53,54,55]. As an example, a prospective study including 185 patients with UC in clinical remission showed that a cutoff point of ≥135 mc/g was able to predict the histological activity of disease (Geboes score > 3.1) with a sensitivity of 54% and specificity of 69% [37].

A systemic review identified FC cutoff points, ranging from 40 to 250 μg/g, to distinguish histological remission from histological activity, indicating that patients with values below the limit had a high rate of histologic remission [56]. A very recent post hoc analysis, including 639 patients with mild to moderate UC (the MOMENTUM trial), found that the optimal cutoff of FC is between 75 and 100 μg/g for the identification of patients with histologic remission [57].

Lastly, in a meta-analysis including nine studies and 1039 patients, an FC cutoff of 100–200 μg/g was reported to have a good diagnostic accuracy to identify patients in histological response. In this study, the pooled sensitivity and specificity were 69% and 77%, respectively [58].

Although changes in FC demonstrate good sensitivity and specificity for assessing disease activity, clinicians should be aware that elevated FC levels may be found in several non-IBD conditions such as infectious enterocolitis and colonic polyps, including inflammatory polyps [59,60]. As a practical conclusion, we propose an algorithm for the use of FC in patients with UC [59] (Figure 1).

In summary, FC but not CRP has a strong correlation with endoscopic and histological activity and predicted clinical relapse. A combination of FC and PRO2 (clinical) scores was shown to be even more accurate. However, the optimal cutoff is yet to be determined, and different cutoffs may be needed to identify endoscopic, histologic activity, and relapse risk. In addition, although CRP is not a sensitive marker in UC in general, together with the clinical scores, it may help in predicting outcomes in patients with ASUC.

## 4. Endoscopic Targets

Although clinical symptoms correlate relatively well with the endoscopic severity in UC, the resolution of symptoms alone is not a sufficient target. Objective evaluation of the mucosal inflammation is necessary when making clinical decisions [9,10]. Approximately 40% of patients in clinical remission have some degree of endoscopic inflammation [27].

In patients with clinical remission, endoscopic mucosal healing is an essential objective target according to the recent STRIDE I-II and the IOIBD consensus [10,61]. Endoscopic healing (EH), defined as MES ≤ 1 or UCEIS 0–1, was associated with improved long-term clinical remission, decreased corticosteroid use, and a need for colectomy and hospitalization [62,63,64].

The Mayo endoscopic score is the most commonly used to assess the endoscopic response and remission in clinical trials, and it has become a widely accepted index for mucosal healing in clinical practice [61]. In clinical trials of infliximab, ACT-1, and ACT-2 in patients with moderate to severe UC, a lower MES at week 8 was also associated with a greater chance of being in clinical remission at week 54 (MES of 0, 47.0%; 1, 35.0%; 2, 5.3%; 3, 5.3%) [65]. Similarly, MES ≤ 1 was used to define an endoscopic mucosal remission in the clinical trials of adalimumab (ULTRA 1–2 studies) and vedolizumab (GEMINI study) in patients with moderate to severe UC [66,67]. In the clinical trial of tofacitinib, a more stringent score, MES of 0, was used for clinical remission [68].

A recent study reported that patients with active UC in a treat-to-target strategy who achieved endoscopic healing (MES 0–1) over two consecutive endoscopies (interval 16 months) have a low risk of relapse. The 1 year cumulative risk of relapse in patients with persistent EH was 11.5% for 26 months [69].

As a practical recommendation, an MES of 1 should probably be a minimum target in endoscopic healing. The current evidence supports that complete endoscopic remission, with MES of 0, is the optimal target for deep remission. Patients who achieved an MES of 0 are more likely to have a longer duration of clinical remission than patients without mucosal healing [10].

Moreover, an MES of 1 is associated with an increased risk of disease recurrence compared to an MES of 0 [70,71,72]. In a prospective study including 187 patients with MES 0 and 1, patients with MES 1 had a significantly higher relapse rate than patients with MES 0 during the first 6 months of follow-up (9.4% and 36.6%, respectively). An MES of 1 was the only factor independently associated with UC relapses (OR 6.27, 95% CI 2.73–14.40) [70]. Another study included UC patients in steroid-free remission, and clinical relapse was significantly more frequent in patients with MES 1 than MES 0 (27.3 vs. 11.5%) for 1 year [73]. The latter study showed a benefit of therapeutic treatment for patients with UC, with an MES of 1 associated with lower rates of clinical relapse and endoscopic exacerbation [72].

More recent studies suggested that patients with UC who achieve completed endoscopic remission (MES 0 or UCEIS 0) have better outcomes and a lower risk of clinical relapse and disease-related complications than those who achieve conventionally defined remission (MES ≤ 1) [74], supporting the use of endoscopic MES 0 as the most suitable treatment endpoint to determine mucosal healing in patients with UC [70,73,75].

In a recent meta-analysis of 17 studies including 2608 patients with UC in clinical remission, patients achieving MES 0 had a 52% lower risk of clinical relapse compared to patients achieving MES 1 (RR 0.48; 95% CI 0.37–0.62). The median 12 month risk of clinical relapse in patients with MES 1 was 28.7%; the estimated annual risk of clinical relapse in patients with MES 0 was 13.7% (95% CI 10.6–17.9) [76].

Multiple endoscopic scores are available in UC, but the MES and Ulcerative Colitis Endoscopic Index of Severity (UCEIS) are the most extensively studied [77]. Importantly, two studies reported a strong correlation between UCEIS and MES for predicting outcome in UC [78,79]. STRIDE I/II recommended the Mayo score for real-world endoscopic healing evaluations, which is easy to apply to clinical trials and clinical practice. However, more recent evidence supports UCEIS as a more extensively validated score with endoscopic remission [61,80] and a better correlation with disease severity and treatment responsiveness than the Mayo score [79,81]. In 2017, the IOIBD suggested the use of UCEIS 0 for the definition of endoscopic remission and the use of a decrease in Mayo endoscopic score ≥ 1 or a decrease in UCEIS ≥ 2 for the definition of endoscopic response in ulcerative colitis, at least for clinical trials based on systematic review and a two-round Delphi consensus [61].

It is important for clinical practice that the UCEIS is more responsive and granular in the assessment of the mucosal status; for example, a study reported that UCEIS more accurately reflected UC severity and clinical outcomes compared to the MES and was more sensitive to detect changes in the mucosal ulceration severity, which the original Mayo score completely overlooks [81]. Mari Arai et al. reported that the duration of recurrence was significantly prolonged in patients with a UCEIS ≤ 1 compared with a UCEIS > 1 (35.9 months vs. 29.0 months, *p* = 0.006). The recurrence rate increased gradually with the UCEIS score (5.0% for UCEIS = 0, 22.4% for UCEIS = 1, 27.0% for UCEIS = 2, 35.7% for UCEIS = 3 and 75.0% for UCEIS = 4–5) in patients with clinical remission [79].

Although the optimal time for assessing endoscopic MH after the initiation of treatment has not been well established, it has been recommended that endoscopic assessment should ideally be performed at 3–6 months after starting therapy in UC patients to decide on further treatment changes [10], Although, in clinical trials, the usual reassessment of endoscopic activity is after 8–12 weeks after the initiation of the therapy, a shorter interval to assess mucosal healing is certainly more feasible in UC compared to CD; yet, the 3–6 month interval was also suggested for practical reasons to be more permissive in an everyday clinical setting and less burdensome for the patient. To support the above interval, a study using a proactive monitoring strategy and assessing the mucosal inflammation within 6 months of initiation of a biologic therapy was associated with a reduction in steroid use in UC [82]. However, in patients with mild disease severity, who do not require biological treatment (or immunosuppressants), the risk of relapse and disease progression is lower, and endoscopic reassessment can be performed after 6 months or when a patient has active symptoms/abnormal biomarkers, more as a point-of-care approach [83].

In conclusion, an MES of 1 should be the minimum endoscopic target in UC. However, recent evidence suggests that more stringent endoscopic goals (Mayo or UCEIS score of 0) can be associated with superior outcomes and lower risk of relapse in the long term. The optimal monitoring frequency should be adjusted to the overall patient severity and earlier disease course; however, an objective assessment of the mucosal healing can be generally suggested after 3–6 months of the change of the therapy, especially if this resulted in a change of the therapeutic approach (e.g., initiation of full dose of steroids, immunomodulators, or biological therapy).

## 5. Histological Targets

There is still debate over whether the histologic target in patients with UC in clinical and endoscopic remission should be considered as an additional treatment endpoint. Despite this, histology was not recommended as a treatment target in the current STRIDE II guideline. However, recently, histological remission (HR) has been an emerging goal and has been recognized as an important prognostic factor and potential treatment target in patients with UC.

Although more than 20 histologic scoring systems are available for assessment of histological disease activity in UC, only three scoring systems have been validated and most used in clinical studies: the Geboes score (GS), Nancy histological index (NHI), and Robarts histopathology index (RHI) [84,85]. All these histological indices are reliable for measuring the disease activity and evaluating response to treatment in UC. NHI and RHI are the two most validated scores with good intra- and interobserver reliability, with comparable results. Although GS is the most used score, it is not fully validated. However, a recent study showed that GS and NHI are strongly correlated (correlation coefficient: 0.882, *p* < 0.001), indicating high concordance with histological remission and response in a patient with UC [86].

In histological healing, normalization of active histological inflammation can be observed, including the absence of basal plasmacytosis, neutrophilic infiltration, and crypt architectural irregularities. These histologic markers of active inflammation should be recognized as a therapeutic target in histological assessment. Quiescent disease is characterized by the absence of mucosal neutrophils, although degrees of architectural changes and chronic inflammation may remain [87]. However, there is no standard definition of histologic remission in clinical practice for UC. Various definitions have been proposed since residual inflammation with architectural distortion to complete normalization of the colonic mucosa [88]. In the proposed definition, HR is characterized by resolving the crypt architectural distortion and the inflammatory infiltrate [89].

According to the histologic indices, the last proposed HR definition is an RHI ≤ 3 with sub-scores of 0 for lamina propria neutrophils and neutrophils in the epithelium and without ulcers or erosions, an NHI grade 0 or 1 (0, absence of significant histological disease; 1, chronic inflammatory infiltrate with no acute inflammatory infiltrate) or a GS < 2A (architectural changes with or without chronic inflammatory infiltrate, with no acute inflammation) [88]. A summary of scoring for the histologic target [84,88] is shown in Table 2.

The added benefit of histological remission (HR) over endoscopic mucosal healing has been shown in multiple studies in predicting better long-term clinical outcomes and the prevention of disease complications. In a meta-analysis of 1573 UC patients, HR was superior to endoscopic and clinical remission in predicting clinical outcomes. HR had a 52% risk reduction in clinical relapse/exacerbation compared to histological activity (RR 0.48, 95% IC 0.39–0.60) [90]. Two studies have shown that HR also predicts corticosteroid use and hospitalization (OR 0.26, 95% CI 0.11–0.56, *p* = 0.002 for steroid use and OR 0.30, 95% CI 0.12–0.76 *p* = 0.01 for hospitalization) [62,91,92].

The latter study evaluated HR in 270 active UC patients with a treat-to-target strategy, who initially achieved persistent EH (MES 0–1) in serial two endoscopies for 16 months. This study showed that HR was associated with a lower risk of clinical relapse (1 year cumulative risk: 18.7% vs. 29.5%; adjusted HR, 0.56 (0.37–0.85)) and a lower risk of hospitalization within 28 months of follow-up (HR 0.44 (0.20–0.94)) [93].

In addition, several studies showed that active histologic activity was an independent risk factor for clinical relapse in patients in achieving clinical and endoscopic remission [71,94,95,96,97,98], using NHI >2 (HR 3.7 (95% CI 1.1–12.3) [71] and GS ≥ 3.1 (HR 3.5, 95% CI 1.9–6.4, *p* < 0.0001) for histologic activity [94].

The real question, however, is whether histology can further differentiate patients in clinical and endoscopic remission. In a recent retrospective study including 269 UC patients in endoscopic remission with an MES of 0, the presence or absence of histologic activity was not associated with time to relapse [99]. In contrast, Gupta et al. reported in a meta-analysis including 28 studies and 2677 patients with UC in endoscopic remission (MES 0–1) that persistent histologic activity is associated with higher rates of relapse (OR 2.41, 95% CI 1.91–3.04), with a similar effect noted in the subgroup with endoscopic MES of 0 vs. 0 or 1 [100]. In a recent meta-analysis, histological healing in patients with clinical and complete endoscopic remission (MES 0) was evaluated in 10 studies. Patients who achieved HR had a 63% lower risk of clinical relapse compared to patients with persistent histologic activity (RR 0.37; 95% CI 0.24–0.56). The estimated annual risk of clinical relapse in those who achieved histologic remission was 5.0% (95% CI, 3.3–7.7) [76].

A more recent prospective study evaluated the benefit of achieving complete histologic normalization (GS of 0) in 83 patients with complete EH (MES 0). Patients with complete histologic normalization had a lower relapse rate compared to those without normalization during the following 2 years (12% vs. 50%, *p* < 0.001) (OR 7.22, 95% CI 2.48–24.70) [101]. A further study reported that only complete histological normalization of the entire colon was associated with improved relapse-free survival (HR 0.23; 95% CI 0.08–0.68), Whereas segmental normalization did not signal improved clinical outcomes [102].

Another angle is the evaluation of specific features; for example, in one of the early studies evaluating histological markers for active disease, the presence of basal plasmacytosis (BPC) was independently related to a higher risk of clinical relapse in patients with clinical remission, with an HR of 4.5–5.1 (95% CI 1.2–1.7 to 11.9–19.9) [98,103]. This result was confirmed by a more recent study; in 76 UC patients with endoscopically quiescent disease, presence of BPC and active histological inflammation (GS ≥ 3.2, OR 8.29 (95% CI 2.49–27.61)) were adjunctive histological markers associated with increased likelihood of disease relapse at 18 months follow-up [104].

A more recent meta-analysis by Gupta et al. evaluated the impact of different histological scales and histologic features, showing that more rigorous Geboes cutoffs demonstrated numerically stronger impact on relapse rates: Geboes < 3.1 (OR 2.40, 95% CI 1.57—3.65), Geboes < 2.1 (OR 3.91, 95% CI 2.21—6.91), and Geboes 0 (OR 7.40, 95% CI 2.00—18.27). Among individual histologic features, basal plasmacytosis (OR 1.94), neutrophilic infiltrations (OR 2.30), mucin depletion (OR 2.05), and crypt architectural irregularities (OR 2.22) predicted relapse of UC. This study showed that a greater degree of histological normalization may have a stronger impact on disease outcomes [100].

Of note, achieving HR was also associated with a reduction in colorectal cancer risk in UC, as shown in a meta-analysis where the pooled odds ratio for colorectal neoplasia was 3.5 (95% CI, 2.6–4.8; *p* < 0.001) in those with any mucosal inflammation and 2.6 (95% CI, 1.5–4.5; *p* = 0.01) in those with histologic inflammation when compared to those with mucosal healing [105].

Despite not currently being recommended as a target by the STRIDE II, HR could be considered as an adjuvant goal to optimize outcomes in UC. For example, achieving HR might be needed before considering stopping or deescalating therapy. Nevertheless, its benefit to everyday clinical practice over endoscopic healing is still unclear in unselected patients. In addition, the definition of HR needs further validation [10]. A summary of selected recent studies is provided in Table 3.

In summary, recent evidence shows that histology can identify a subset of patients in clinical and endoscopic remission that are still at higher risks for clinical relapse; accordingly, it can be regarded as a more stringent treatment endpoint. However, in everyday clinical practice, it remains unclear if histological healing should be used as a routine therapeutic target in addition to clinical, biomarker, and endoscopic assessment in unselected UC patients.

## 6. A Tailored Approach to Treatment Target(s) in UC

A tailored approach to the treatment targets according to disease severity and risk factors may determine the success of treatment. In UC, disease severity, extent, and progression are variable. Thus, patient characteristics (and monitoring needs) may be significantly different in patients presenting with ASUC or late-onset mild UC of a limited extent. Knowing the earlier disease course and identifying predictors of disabling disease are important to initiate a personalized T2T therapeutic strategy, i.e., “the right target to the right patient” according to risk stratification in the given patient [106].

Patients with high risk for aggressive disease course and colectomy may include patients with young age at diagnosis (<40 years old), extensive colitis, severe endoscopic activity, presence of extraintestinal manifestations, early need for corticosteroids, and elevated inflammatory markers [107]. These patients may benefit from intensive therapy from the beginning to prevent the disease progression and consequent disease complications [108]. Therefore, tight monitoring of the targets is recommended in patients with a high disease burden. On the other hand, step-up therapy is still suitable in low-risk patients to avoid overtreatment in patients who are likely to have a benign course of the disease. The physician should balance the benefits and risks and set targets for each individual.

In the early stage of disease, we should aim high, including clinical, biomarker, and endoscopic (histologic?) healing. Conversely, the target may be different in patients with later-stage/burned-out disease (tube-like colon). The control of clinical symptoms with minimal inflammation may not achieve complete endoscopic or histologic remission. The proposed practical algorithm for a personalized “treat-to-target” approach in UC is shown in Figure 2.

## 7. Conclusions

Implementation of a treat-to-target strategy in everyday clinical practice remains challenging in UC. Clinical remission (resolution of rectal bleeding and normalization of bowel habits) and endoscopic remission (MES ≤ 1) can be suggested as the minimal recommended targets for everyday practice in concordance with the updated STRIDE consensus. However, current evidence supports complete endoscopic remission (MES or UCEIS of 0) as the optimal target for mucosal healing. In addition, FC has a good correlation with endoscopic and histologic activity, and a combination of FC and clinical score (PRO2) can guide the clinician to prioritize patients that warrant endoscopy for disease reassessment.

Many recent studies highlighted that histological activity better predicts long-term disease outcomes and mucosal healing in UC. However, none of the available histologic scores can be considered as the gold standard, and sampling bias/variation may also apply.

Therefore, it remains questionable if histological healing should be used as a routine therapeutic endpoint in addition to clinical, biomarker, and endoscopic targets. In addition, cross-sectional imaging modalities, including ultrasonography (US), computed tomography (CT), and magnetic resonance imaging (MRI), are not currently recommended as the treatment target in UC [10]. However, bowel US has a good correlation with endoscopic activity and can be used at the point of care in assessing the disease activity and severity of mucosal inflammation UC [109,110,111].

Importantly, none of the treatment targets can be used as a sole parameter. While clinical symptoms and biomarkers are usually regarded as short-term markers, flare frequency and biomarker patterns can also be helpful to estimate longer-term disease course in the given patient. Thus, understanding the disease course and stratifying the risk factors are key components to establish the optimal target for each patient to balance the risk–benefit and cost-effectiveness of the treat-to-target strategy in clinical practice.

## Figures and Tables

**Figure 1 jcm-10-05551-f001:**
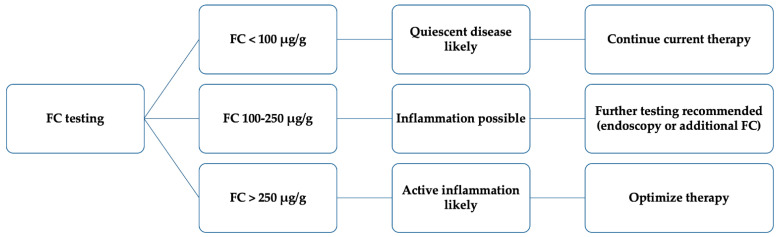
Interpreting fecal calprotectin (FC) testing; modified from Bressler et al. [59].

**Figure 2 jcm-10-05551-f002:**
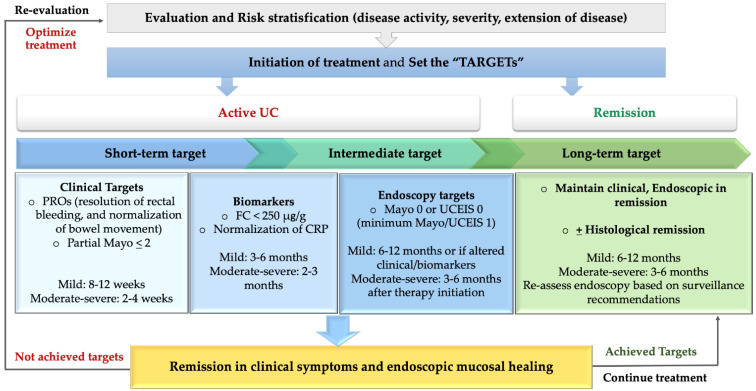
Proposed algorithm for “treat-to-target” approach in ulcerative colitis.

**Table 1 jcm-10-05551-t001:** Summary of proposed recommended targets of “treat-to-target” approach in UC management.

Treatment Target	Targets to Achieve	Time to Reassess Targets	
		Clinical Active	Clinical Remission
1. Clinical symptoms	PRO2 remission (RB = 0 and SF ≤ 1) Partial Mayo score (<2)	Mild: 8–12 weeks Moderate–severe: 2–4 weeks	Mild: 6–12 months Moderate–severe: 3–6 months
2. Biomarkers	CRP < normal limit FC < 250 μg/g (minimum), <100 μg/g (optimum)	Mild: 3–6 months Moderate–severe: 8–12 weeks after therapy initiation	Mild: 6–12 months Moderate–severe: 3–6 months
3. Endoscopy (mucosal healing)	MES 0 (1, minimum) UCEIS 0 (1, minimum)	Mild: 6–12 months or if altered symptoms or abnormal biomarkers Moderate–severe: 3–6 months after therapy initiation (STRIDEI, II)	Based on screening recommendations in deep remission Prompted by clinical symptoms or (consecutive) biomarker positivity
4. Histology (histological remission/ response)	Nancy histological index (NHI) Geboes score (GS) Robarts histopathology index (RH) (see Table 2)	Adjunctive (added-on) target	Adjunctive (added-on) target

**Table 2 jcm-10-05551-t002:** Summary of the scoring system in histologic target in UC.

Histological Targets	Scoring Systems		
	Geboes Score (GS)	Robarts Histopathology Index (RHI)	Nancy Histological Index (NHI)
Remission	GS < 2A	RHI ≤ 3: without neutrophils in lamina propria and epithelium and without ulcers or erosions	Grade 0
Response	GS ≤ 3	RHI ≤ 9: without neutrophils in the epithelium and without erosions or ulcers	Grade 1
Active disease activity	GS > 3	Not clearly defined	Grade ≥ 2

**Table 3 jcm-10-05551-t003:** The selected current studies of clinical, biomarker, endoscopic, and histological targets in the treatment of patients with UC.

Study (Year)	Study Type and Population	Treatment Targets	Main Results and Outcomes
**1. Clinical target**			
S. Restellini et al. (2019) [24]	Systematic review of 23 studies (*n* = 3320)	PROs vs. endoscopic activity	Composite clinical measures including rectal bleeding (RB) and stool frequency (SF) had moderate to strong correlations with endoscopic disease activity; absence of RB identified patients with an inactive disease with higher levels of sensitivity than normalization of SF
N. Narula et al. (2019) [25]	Meta-analysis of 5 studies (*n* = 2132)	PROs vs. endoscopic remission (MES 0–1)	Combined 2 items, RB and SF subscores of 0 (Se 36%, Sp 96%, pos-LR 8.4, neg-LR 0.66 for endoscopic remission); RB subscore of 0 (Se 81%, Sp 68%, pos-LR 2.5, negative LR 0.28); SF subscore of 0 (Se 40% Sp 93%, pos-LR 6.0, and neg-LR 0.64)
P.A. Golovics (2021) [26]	Prospective study (*n* = 171)	Clinical scores (PRO2, partial Mayo, SCCAI) vs. endoscopic scores (MES, UCEIS)	RB, SF subscore of 0, or PRO2 remission (RBS0 and SF ≤ 1), partial Mayo (≤2), and SCCAI (≤2.5) remission were similarly associated with MES ≤ 1 or UCEIS ≤ 3 scores in ROC analysis (AUC: 0.93–0.72)
**2. Biochemical target**			
L. Hart et al. (2020) [37]	Prospective study (*n* = 185)	FC cutoff level vs. endoscopic and histologic activity	FC ≥ 170 μg/g predicts active endoscopic activity (MES 2–3 from MES 0–1) (Se 64%, Sp 74%), and FC ≥ 135 μg/g predicts active histological activity (Se 54%, Sp 69%)
X. Ye et al. (2021) [58]	Meta-analysis of 9 studies (*n* = 1039)	FC vs. histologic response and remission	Accuracy of FC for histological remission: Se, Sp, and AUC of 76%, 71%, and 79%, respectively; accuracy of FC for histological response: Se Sp, and AUC of 69%, 77%, and 80%, respectively.
P.S. Dulai (2020) [45]	Systemic review of 26 studies (*n* = 2886)	Combined FC cutoff level and PRO2 vs. endoscopic activity	PRO2 remission (RBS 0 + SFS 0/1) and FC ≤ 50μg/g may avoid endoscopy in 50% patients with a false-negative rate < 5% RBS 2/3 + SFS 2/3 and FC ≥ 250 μg/g may avoid endoscopy in approximately 50% of patients with false-positive rate < 5% RBS 0 but SFS 2/3 led to false-negative and false-positive rates consistently > 10%, and endoscopic evaluation may be warranted
J. Li et al. (2019) [47]	Meta-analysis of 14 studies (*n* = 1110)	FC cutoff level vs. clinical relapse	9 studies used FC cutoff ≥ 150 μg/g: Se71% and Sp 86%; 5 studies used FC cutoff < 150 μg/g: Se 79% and Sp 64% in predicting relapse; most of the studies had follow-up ≥ 12 months
**3. Endoscopic target**			
M. Arai et al. (2016) [79]	Prospective study (*n* = 285) UC patients with clinical remission	UCEIS vs. MES vs. clinical relapse	UCEIS correlated with MES (*r* = 0.93). The recurrence rate was 5.0% for UCEIS = 0, 22.4% for UCEIS = 1, 27.0% for UCEIS = 2, 35.7% for UCEIS = 3, and 75.0% for UCEIS = 4–5 during follow-up of 48 months
P. Boal Carvalho et al. (2016) [73]	Retrospective UC with corticosteroid-free remission (*n* = 138)	MES 0 vs. MES 1 vs. clinical relapse	Clinical relapse more frequent in patients with MES 1 than MES 0 (27.3 vs. 11.5%, *p* = 0.022); MES 1 increased risk of relapse (OR 2.89, 95% CI 1.14–7.36, *p* = 0.026) during 12 months
M. Barreiro-de Acosta et al. (2016) [70]	Prospective UC (*n* = 187)	MES 0 vs. 1 vs. clinical relapse	The relapse rates in patients with Mayo scores 0 and 1 were 9.4% and 36.6%, respectively, during the first 6 months
S. Jangi et al. (2020) [93]	Retrospective UC patients (*n* = 270) with persistent EH (two serial endoscopies)	MES 0 vs. MES 1 vs. clinical relapse	1 year CR of relapse in patients with persistent EH was 11.5% and in patients with persistent histological remission was 9.5% (interval of EH evaluation: 16 months)
H. Yoon et al. (2020) [76]	Meta-analysis of 17 studies (*n* = 2608) UC patients in clinical remission	MES 1 vs. MES 0 vs. clinical relapse	MES 0: 52% lower risk of CR (RR 0.48; 95% CI, 0.37–0.62) The median 12-month risk of CR in patients with MES 1 was 28.7%; the estimated annual risk of CR in MES 0 was 13.7% (95% CI, 10.6–17.9)
**4. Histological target**			
S. Park et al. (2016) [90]	Meta-analysis of 13 studies (*n* = 1360)	Histologic activity vs. clinical relapse	In patients with clinical and endoscopic remission, HR was associated with lower clinical relapse (CR), RR 0.48, 95% CI: 0.39–0.60 during follow-up of 12 months
R.K. Pai et al. (2020) [92]	Prospective study (*n* = 281)	Histologic activity (GS ≥ 2B.1) vs. need for corticosteroids	The histologic activity was associated with systemic corticosteroid use (OR 6.34; 95% CI, 2.20–18.28; *p* = 0.001); mucosal neutrophils had higher rates of corticosteroid use (*p* < 0.001) during follow-up of 3 years
K.C. Cushing (2020) [76]	Prospective study (*n* = 83) UC patients with MES 0	Complete histologic normalization (Geboes score = 0) vs. relapse	Patients with complete histologic normalizations were less likely to have relapse compared to those without normalization (12% vs. 50%, *p* < 0.001) (OR 7.22, 95% CI 2.48–24.70) during follow-up of 2 years
B. Christensen et al. (2020) [101]	Retrospective study (*n* = 646) UC patients	Segmental vs. complete colon histological normalization vs. clinical relapse	Complete histological normalization of the bowel was associated with improved relapse-free survival (HR 0.23; 95% CI 0.08–0.68; *p* = 0.008); segmental normalization did not improve clinical outcomes
A. Gupta et al. (2020) [100]	Meta-analysis of 28 studies (*n* = 2677) UC patient with MES 0–1.	Histologic activity vs. clinical relapse	Histologically active increased risk of relapse (OR 2.41, 95% CI 1.91–3.04), basal plasmacytosis (OR 1.94), neutrophilic infiltrations (OR 2.30), mucin depletion (OR 2.05), and crypt architectural irregularities (OR 2.22) during follow-up of 12–72 months
H. Yoon et al. (2020) [76]	Meta-analysis of 10 studies, UC patients with MES 0	Histologic activity vs. clinical relapse	HR had a 63% lower risk of relapse vs. patients with persistent histologic activity (RR, 0.37; 95% CI, 0.24–0.56); the estimated annual risk of clinical relapse in HR was 5.0% (95% CI, 3.3–7.7%)

## Data Availability

Not applicable.

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
