# Peer review of "Treatment Targets in Ulcerative Colitis: Is It Time for All In, including Histology?"

_jcm, 2021, doi:10.3390/jcm10235551_

Round 1

Reviewer 1 Report

Thank you for this manuscript. I find it well written and most importantly, clinically useful. It does not provide insights on new potential biomarkers in the field, which would add more scientific value, however, the clinical value is important and I find this as an advantage. Moreover, data are clearly presented which makes it easy to follow. 

Author Response

Reviewer 1

Point 1

Thank you for this manuscript. I find it well written and most importantly, clinically useful. It does not provide insights on new potential biomarkers in the field, which would add more scientific value, however, the clinical value is important and I find this as an advantage. Moreover, data are clearly presented which makes it easy to follow. 

Response 1

Thank you. We appreciate your valuable suggestion. We recognized the importance of the new potential biomarkers. However, in this review, we focused on the current treatment target in everyday clinical practice based on STRIDE I-II consensus guidance. 

Reviewer 2 Report

In this review, Panu Wetwittayakhlang, et al. discussed the current evidence for the different treatment targets and their value in everyday clinical practice.

Strengths of this study:

Authors have done a good job in delivering a meticulous summary of all the existing research in response to research question.

Minor concerns:

  • Authors may want to comment on the imaging targets like MRE and US, and how are they comparable to endoscopic assessment and mucosal healing.
  • There are some minor English language errors (For example, “In addition, clinical scores have a good correlation with endoscopic activity, which can use to prioritize patients for endoscopic evaluation” rather than “In addition, clinical scores have a good correlation with endoscopic activity, which can be used to prioritize patients for endoscopic evaluation”). Consider careful revision with attention to grammar to improve readability.
  • Authors may want to add strengths and limitations of this review. There are several articles addressing the same issue.

Author Response

Point 1:  Authors may want to comment on the imaging targets like MRE and US, and how are they comparable to endoscopic assessment and mucosal healing.

Response 1:  Thank you for your valuable suggestions. We added the usefulness of imaging assessment including US, CT, and MRI on the updated data in the conclusion of the manuscript as following;

In addition, cross-sectional imaging modalities, including ultrasonography (US), computed tomography (CT), and magnetic resonance imaging (MRI), are not recommended as the treatment target in UC at the moment. However, bowel US has a good correlation with endoscopic activity and can be used at a point-of-care in assessing the disease activity and severity of mucosal inflammation UC.

Point 2

There are some minor English language errors (For example, “In addition, clinical scores have a good correlation with endoscopic activity, which can use to prioritize patients for endoscopic evaluation” rather than “In addition, clinical scores have a good correlation with endoscopic activity, which can be used to prioritize patients for endoscopic evaluation”). Consider careful revision with attention to grammar to improve readability.

Response 2  Thank you, we edited the English error as suggested.

Point 3

Authors may want to add strengths and limitations of this review. There are several articles addressing the same issue.

Response 3 We highlighted the strengths of this review in the introduction part the manuscript, as following; We highlight the important state-of-the-art care and practical up-to-date evidences in the different treatment targets.”

Reviewer 3 Report

The authors present a review of the different treatment targets for ulcerative colitis (UC). UC is a gastrointestinal disorder caused by chronic inflammation. The current methods of assessment for optimal treatment is complex and multi-faceted. 

The review is well written and enjoyable to read. It covers the use of clinical scores, patient reported outcomes and biomarkers. The authors argue the case for the application of histological target as another predictor of recurrence. 

The existing literatures are extensively researched and presented concisely to the readers. This review would be a useful resource for clinicians managing UC and provides a balanced view of the topic. 

Minor amendments: 

Figures 1 & 2 appear blurry, please update to higher resolution images. 

Table 3: center "1. Clinical targets" 

Author Response

Point 1

Figures 1 & 2 appear blurry, please update to higher resolution images. 

Table 3: center "1. Clinical targets" 

Response  1 Thank you for your suggestion and comment. Figure 1 and Figure 2 were updated to higher resolution, and in the table 3 “Clinical targets” was centralized.